# Local Pro- and Anti-Coagulation Therapy in the Plastic Surgical Patient: A Literature Review of the Evidence and Clinical Applications

**DOI:** 10.3390/medicina55050208

**Published:** 2019-05-24

**Authors:** Jeremie D. Oliver, Emma P. DeLoughery, Nikita Gupta, Daniel Boczar, Andrea Sisti, Maria T. Huayllani, David J. Restrepo, Michael S. Hu, Antonio J. Forte

**Affiliations:** 1Mayo Clinic School of Medicine, Mayo Clinic, Rochester, MN 55905, and Scottsdale, AZ 85259, USA; oliver.jeremie@mayo.edu (J.D.O.); deloughery.emma@mayo.edu (E.P.D.); gupta.nikita@mayo.edu (N.G.); 2Division of Plastic Surgery and Robert D. and Patricia E. Kern Center for the Science of Health Care Delivery, Mayo Clinic, Jacksonville, FL 32224, USA; danielboczar92@gmail.com (D.B.); asisti6@gmail.com (A.S.); maria.t.huayllanip@gmail.com (M.T.H.); rpo20@hotmail.com (D.J.R.); 3Department of Plastic Surgery, University of Pittsburgh Medical Center, Pittsburgh, PA 15213, USA; hums2@upmc.edu

**Keywords:** coagulation, anti-coagulation, drug delivery, local therapy, flaps, plastic surgery

## Abstract

The risks of systemic anti-coagulation or its reversal are well known but accepted as necessary under certain circumstances. However, particularly in the plastic surgical patient, systemic alteration to hemostasis is often unnecessary when local therapy could provide the needed adjustments. The aim of this review was to provide a summarized overview of the clinical applications of topical anti- and pro-coagulant therapy in plastic and reconstructive surgery. While not a robust field as of yet, local tranexamic acid (TXA) has shown promise in achieving hemostasis under various circumstances, hemostats are widely used to halt bleeding, and local anticoagulants such as heparin can improve flap survival. The main challenge to the advancement of local therapy is drug delivery. However, with increasingly promising innovations underway, the field will hopefully expand to the betterment of patient care.

## 1. Introduction

When the homeostasis of hemostasis is disrupted, potentially devastating thrombosis or hemorrhage can result. Various therapies exist to manipulate the components of the coagulation system to achieve the desired result, but these are often administered on a systemic scale. For the surgical patient in need of pro- or anti-coagulation, local agents that could accomplish the needed alterations in coagulation without the risks of a systemic therapy would be ideal. Here, we discuss situations in which local alteration of hemostasis would be useful in the plastic surgery patient, as well as the available pharmacologic therapies.

A wide range of local pro- and anti-coagulation options exist in the literature, with the earliest documented medical records of ancient Egypt, Greece, and Native America citing the use of agents common in nature such as wax, grease, barley, and animal hide to achieve hemostasis [1]. In the modern era, advances in topical agents have expanded the possibilities to manipulate hemostasis in the surgical patient. These advances include physical (i.e., bone wax and ostene), absorbable (i.e., oxidized cellulose and gelatin foams), and biologic agents (i.e., topical thrombin, fibrin, platelet gel). Synthetic agents, such as cyanoacrylate and polyethylene glycol, have also emerged over the last decade [1]. In searching for the ideal local agent, factors such as ease of use, delivery mechanisms, efficacy, lack of antigenic properties, absorption, and cost are all of great relevance. While no single local agent developed to date has maximized all these qualities, the development of local coagulation agents continues to move forward to optimize efficacy and cost and, ultimately, the greatest outcome for the patients. The aim of this review was to provide a summarized overview of the clinical applications of topical anti- and pro-coagulant therapies relevant to plastic and reconstructive surgery.

### 1.1. Pro-Coagulants

Minimizing blood loss during and after surgery is critical, and many modalities exist to decrease intraoperative and postoperative bleeding. However, under certain circumstances, a systemic therapy may be undesirable, such as in the patient with a superficial injury who is anticoagulated due to a high risk for thrombotic events. Given the substantial risk of intraoperative and postoperative bleeding in such patients, management often involves the temporary pre- and post-operative interruption of the anticoagulant and (particularly in the case of vitamin K antagonists) the use of short-acting agents, such as unfractionated heparin or low-molecular-weight heparin (LMWH). Known as bridging anticoagulation, this strategy is meant to aid in achieving a normal or near-normal hemostasis at the time of surgery, while still maintaining a low risk of thromboembolism. However, this is still a departure from the patient’s standard therapy, and for particularly high-risk patients, any interruption in their anticoagulation treatment could precipitate a thrombotic event. Conversely, too short a time between the last dose of anticoagulation and surgery or between surgery and resumption of anticoagulation, could result in hemorrhage and hematoma formation, potentially compromising the results of the surgery or causing new complications. The use of LMWH is associated with an increased risk of postoperative wound hematoma following pacemaker and implantable defibrillator placement [2]. The use of warfarin postoperatively also increases the risk of hematoma formation and infection in total hip arthroplasty [3]. The preoperative use of aspirin or warfarin increases the risk of re-bleeding after surgical evacuation of chronic subdural hematomas [4]. The increasingly used direct oral anticoagulants (DOACs), including the three factor Xa inhibitors (apixaban, edoxaban, and rivaroxaban) and one direct thrombin inhibitor (dabigatran), appear to have similar intraoperative transfusion requirements and post-operative bleeding risks as vitamin K antagonist therapy (warfarin), at least in the orthopedic setting, suggesting that hemorrhage will remain a concern in the anticoagulated patient, even as more patients transition away from warfarin [5]. While a non-pharmacologic option to prevent thromboembolism would seem ideal when considering surgery in the thromboembolism-prone patient, the current options are non-ideal. These options include various inferior vena cava filters that may prevent pulmonary embolism by catching the embolus and preventing it from reaching the lungs. However, these filters carry a high risk of complications, including thrombosis of the filter itself, and they do not prevent the formation of thrombus, merely its most deadly complications [6]. Given the risks of bleeding in the anticoagulated patient and the risks of thrombosis if anticoagulation is stopped, a local agent that either enhances clot formation or impedes clot degradation would be of benefit, allowing the reversal of anticoagulation at a specific site while maintaining anticoagulation systemically. Some of these local options include tranexamic acid (TXA), topical thrombin and fibrin, and hemostats.

Topical TXA may be used to control bleeding following oral surgery, as well as epistaxis secondary to coagulation disorders and anticoagulation [7,8]. An anti-fibrinolytic, TXA binds to plasmin, preventing fibrin’s degradation. In addition to its topical uses, TXA has been shown to reduce postoperative bleeding following cardiac surgery when applied in the pericardial cavity [9]. It also appeared to be effective at reducing gross hematuria during bladder irrigation when applied intravesically, though it was less effective at reducing transfusion requirements [10]. When applied topically or intra-articularly following total knee arthroplasty, TXA reduced both blood loss and transfusions [11,12]. Similar results have been found among other surgical patients, but the topical TXA’s effect on thromboembolic risk remains unknown [13].

Topical thrombin and fibrin formulations are also available to achieve hemostasis, particularly in the setting of dermatologic surgery [14]. Fibrin sealants are also widely used for achieving hemostasis in laparoscopic surgery [15,16]. Fibrin sealants composed of fibrinogen and thrombin are particularly well-suited for anticoagulated patients, as fibrin, being the end product of the coagulation cascade, does not require biologic activation. However, bovine-derived thrombin has been shown to induce the formation of antibodies against thrombin, prothrombin, factor V, and cardiolipin, inducing a potentially life-threatening hemorrhagic tendency [1]. ecombinant human thrombin appears to have similar efficacy and safety without these immunologic side effects.

A wide variety of hemostats are also available to achieve hemostasis, with the composition determined by the surgical need as well as their own advantages and disadvantages [17]. Hemostats are dressings, sponges, meshes, or powders that can be applied to a bleeding area to facilitate coagulation. Physical and absorbable hemostats, such as bone wax, collagen and cellulose, are well-suited to control low-pressure bleeding but may embolize and may interfere with healing [18]. Biologic agents, such as thrombin or fibrin sealants, may elicit an immunologic response and are expensive, but can be applied quickly and exert their effect rapidly. Fibrin sealants are less useful in a trauma setting, however, due to their need for a dry field. They are most useful in the patient with coagulopathy, since, by directly generating fibrin—the end product of the coagulation pathway—the anticoagulant’s effect is bypassed [18]. Less effective in coagulopathic patients are agents that rely on an intact coagulation pathway, such as those containing collagen, cellulose, or gelatin. In general, gelatin-based hemostats have been found to be less effective at achieving hemostasis, while fibrin and p-GlcNac-based hemostats have been found to be more effective [17]. In regard to patient safety, polysaccharide hemostats appear to offer the best safety profile and are also useful in trauma cases. 

Please refer to Table 1 for several references of pro-coagulant TXA in plastic, reconstructive, and craniomaxillofacial surgery [19,20,21], as well as to Table 2, outlining studies analyzing the use of topical thrombin [22,23].

### 1.2. Anticoagulants

Local anticoagulants also have their place, as systemic anticoagulation increases the risk of hemorrhage. Even twice-daily enoxaparin prophylaxis for venous thromboembolism may increase clinically relevant bleeding after plastic and reconstructive surgery [18]. Intraoperatively, the use of topical anticoagulants is associated with increased survival of replanted digits, skin grafts, and flaps during free-tissue transfer [24]. Perhaps the oldest anticoagulant, the leech *Hirudo medicinalis* has proven efficacy in reducing venous congestion of skin flaps in plastic surgery, thereby preventing loss of the flap [25,26]. The secret to the leech’s success is its secretion of hirudin, an inhibitor of thrombin. Complications of leech therapy include excess hemorrhage and infection with *Aeromonas hydrophilia*, a gram-negative bacillus native to the leech’s digestive tract [26]. Hirudin is also available in purified and recombinant forms, and a hirudin-containing cream has been shown to improve mild-to-moderate bruise resolution [27]. See Table 3 for several published studies analyzing the use of *H. medicinalis* in plastic and reconstructive surgery [28,29,30,31].

Likely the most well-known anticoagulant, heparin binds to antithrombin, potentiating and accelerating the inactivation of thrombin and factor Xa. Though often an enemy of the surgeon due to its pro-hemorrhage nature, heparin does have some surgical applications. In microsurgery, local unfractionated heparin was as effective as systemic heparin in reducing arterial thrombus size in a rat model, with less alteration to hemostatic parameters [32]. When injected subcutaneously, LMWH has been shown to improve free and regional flap survival, acting in much the same manner as hirudin by decreasing venous congestion [33]. LMWH can also be used topically to improve the resorption of skin hematomas and has also been shown to reduce pain and inflammation and improve skin healing [34].

One of the most efficacious local anticoagulant agents is tissue factor pathway inhibitor, a naturally occurring protein inhibitor of factor X and the tissue factor-factor VII complex of the extrinsic pathway of coagulation. When compared in a rabbit model of near-total ear avulsion injury, topical tissue factor pathway inhibitor therapy led to significantly higher patency rates than heparin, hirudin, or control solutions [35]. Its application in chronic wound therapy also improved blood flow in the wound bed, thereby promoting healing [36].

Less local than targeted, catheter-directed thrombolysis is an increasingly utilized treatment option for pulmonary emboli and deep-vein thromboses, both common and feared complications of many surgeries. By focusing the anticoagulant at the clot, catheter-directed thrombolysis allows the use of lower doses of anticoagulants, theoretically reducing the risk of bleeding [37]. In intermediate-risk pulmonary emboli, however, catheter-directed thrombolysis appears to carry a slightly higher risk of bleeding, though with no difference in mortality [38]. In regard to massive and sub-massive pulmonary emboli, catheter-directed thrombolysis may have a lower bleeding risk than systemic thrombolysis, with equal or lower mortality [39]. Like with most local coagulation therapies, more study is needed in this field to better identify the situations and patients who would most benefit from catheter-directed thrombolysis.

### 1.3. Drug Delivery

The greatest challenge of any local therapy, including manipulation of coagulation, is adequate drug delivery and targeting, and its overcoming could allow for expanded utility. Catheter-directed thrombolysis is leading the way in this field by directing potent thrombolytics to the desired site. Some novel targeting methods include collagen sponges containing biodegradable thrombin-loaded microspheres and gelatin sponges layered with active coagulation proteins, though elongating shelf life and manufacturing remain difficulties [18]. Balloon catheters have been shown to be capable of delivering both heparin and argatroban into injured iliac arteries in dogs, with high doses of the drugs inhibiting thrombus formation [40].

A large challenge with topical therapy in particular is the delivery of the drug through the skin, which acts as a barrier to both large and charged molecules. The outermost layer of the skin, the stratum corneum, acts as the primary barrier to molecule entry [41]. In order to pass through the skin, most molecules must slide between the lipid bilayers of neighboring corneocytes, though some absorption may also occur through follicles. Given that the stratum corneum is the primary barrier to absorption, the removal of this layer improves drug delivery [41]. Such a solution, however, is unpleasant for the patient, and the elimination of the skin’s natural defense against the environment would increase the risk of infection. A more palatable option is lipid-based vesicles, which envelop drugs in a bubble of variable lipophilicity that more easily penetrates the skin and may also protect the drugs from degradation. Such vesicles can be delivered in a variety of forms, including gels, creams, transdermal patches, and injections [41]. When applied to a field of local coagulation, liposomes appear to improve the delivery of LMWH into the skin [35]. A negatively charged large molecule, LMWH does not penetrate the skin well, but attachment to a positively charged liposome appears to enhance its delivery. As topical drug delivery methods improve, options for local coagulation therapy should expand as well.

### 1.4. Pros and Cons

The main benefit of local therapy is that the systemic effects of altered coagulation are avoided. In the anticoagulated patient, local reversal of anticoagulation during and following a minor procedure could decrease the risk of stroke, deep-vein thrombosis, or other thrombotic complications that could occur if the patient were taken off their anticoagulation regimen. Likewise, systemic anticoagulation carries the risk of potentially fatal hemorrhage, which could be avoided if the patient only needs local anticoagulation for a skin graft or anastomosis.

Besides drug delivery and targeting, difficulties of local therapy include the little evidence that supports their use, with hemostats being a notable exception, except for a few well-defined circumstances. The situations in which such therapy would be beneficial are also currently mostly limited to superficial injuries, though the potential exists for their application to a wide variety of surgeries. Local pro-coagulants, injected like a local anesthetic, could allow for even large surgeries, while maintaining systemic anticoagulation. While local anti-coagulants have their place in improving venous congestion and healing in superficial surgeries, they could also have a role in improving the gastrointestinal anastomosis blood flow or in enhancing the survival of vascular grafts. With alterations to hemostasis, the rewards have always come with heavy risks, but those risks could lessen substantially with the use of local therapy.

## 2. Conclusions

Local pro- or anti-coagulation therapy shows great promise for delivering the needed alterations to hemostasis in the plastic surgical patient without the risks of systemic therapy. Unfortunately, local therapy is still limited in scope and utility, but there are great opportunities for continued research and development in the field. Advancements in topical drug delivery in particular will hopefully allow for the expansion of this field, with subsequent improvement in patient care and outcomes.

## Figures and Tables

**Table 1 medicina-55-00208-t001:** Studies Analyzing the Use of Tranexamic Acid (TXA).

Article	Indication for Use	Dose and Duration of Treatment	Therapeutic and Side Effects
**Cansancao 2018 [19]**	Minimize perioperative blood loss in liposuction	10 mg/kg IV, pre-op and post-op	TXA group had 56.2% less blood loss by volume for every L of aspirate compared to the placebo group
**Zirk 2018 [8]**	Topical application of TXA to stop oral bleeding in an emergency department	Compression with a gauze soaked with TXA	Oral bleeding improved by 1.5 times when gauze was soaked with TXA Moderate and severe bleeding still necessitated sutures/native collagen fleeces
**Kurnik 2017 [20]**	Minimize blood loss and transfusions in patients undergoing open calvarial vault remodeling for craniosyntosis	10 mg/kg loading dose followed by a 5 mg/kg/h infusion for the first 24 h	TXA administration reduced blood loss Patients receiving TXA required less blood transfusion peri-operatively, and no blood transfusions post-operatively No adverse events reported
**Arantes 2017 [21]**	Randomized control trial (RCT) aiming to reduce intraoperative bleeding in palatoplasties in 66 patients	10 mg/kg loading dose, followed by continuous infusion of 1 mg/kg/h of the same until end of surgery	No significant difference in blood loss between intervention and control group No adverse events associated with TXA administration

**Table 2 medicina-55-00208-t002:** Studies Analyzing the Use of Topical Thrombin.

Article	Indication for Use	Dose and Duration of Treatment	Therapeutic and Side Effects
**Bowman 2018 [22]**	Reduce blood loss in primary total hip arthroplasty using thrombin–collagen and autologous platelet-rich plasma		No significant differences in operative blood loss, drain output, or length of hospitalization
**Ofodile 1991 [23]**	Reduce bleeding in 24 burn patients undergoing skin graft		43.5% reduction in bleeding No adverse effect on rate of wound healing

**Table 3 medicina-55-00208-t003:** Studies Analyzing the Use of Medicinal Leeches.

Article	Indication for Use	Dose and Duration of Treatment	Therapeutic and Side Effects
**Karino 2018 [28]**	Treat venous congestion in a patient after free forearm flap reconstruction for oral cancer	Leeches used twice daily for 5 days	Decreased bleeding from skin flap and improved color No dysfunction of flap
**Butt 2016 [29]**	Treat venous congestion in flaps, digital replants, and revascularizations in 18 patients	Duration ranged from 1 to 8 days, approximately every 4 hours	Successful tissue salvage in 65% of cases Rate of tissue salvage was poor in digital replants and free flaps
**Moffat 2015 [30]**	Salvage flap in nipple areolar complex in a patient post-reduction mammoplasty	Applied three times per day for 3 days, while receiving hyperbaric oxygen	Successful salvage of the flap No need for further surgery
**Grobe 2012 [31]**	Treat venous congestion of flaps used in reconstructive maxillofacial surgery in 148 patients	Dependent on size of impaired area	No complications; improvement of graft perfusion Risk of *Aeromonas* infection in immunosuppressed patients 94 patients received additional surgery or revision of anastomosed vessels

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
