# Peer review of "Local Pro- and Anti-Coagulation Therapy in the Plastic Surgical Patient: A Literature Review of the Evidence and Clinical Applications"

_medicina, 2019, doi:10.3390/medicina55050208_

Round 1

Reviewer 1 Report

Manuscript ID: medicina-460093

Title: Local Pro- and Anti-Coagulation Therapy in the Plastic Surgery Patient: A Literature Review of the Evidence and Clinical Application

Authors: Oliver JD, DeLoughery EP, Boczar D, Sisti A, Huayllani MT, Restrepo DJ, Hu MS, Forte AJ.

The above manuscript is interesting and undertakes an important problem of the local pro- and anticoagulative therapy in plastic and reconstructive surgery. However, the manuscript exhibits some shortcomings and for this reason the current version of the manuscript cannot be accepted for publication.

Major point:

1.      The manuscript should be more practical and interesting for researchers and surgeons. For this reason, the authors should prepare separate tables for each, method or medication presented in the manuscript. Moreover, experimental studies on animals and clinical observation should be also, presented in separate tables. Tables should present data shown in published articles representative for each method. Tables should contain the name of the first author, year of publication, item number in References, indications and contraindications for presented method, therapeutic effects, doses and duration of treatment, side effects and their frequency, as well as prophylaxis to prevent side effects. In the text of the manuscript, the authors should discuss and conclude data presented in tables.

2.      Moreover, the authors should present/calculate average cost of each method per one patient.

Minor points

1.      Page 2, line 67. “The increasingly used direct oral anticoagulants appear to have similar intraoperative transfusion requirements and post-operative bleeding risks as warfarin…” What does this sentence mean? Warfarin is the oral anticoagulant, too. This sentence should be corrected.

2.      Page 2, line 84-86. “It also appears to be effective at reducing serum volume for bladder irrigation when applied intravesically for hematuria…” What is the meaning of this sentence? It should be clarified.

3.       Page 5, References, item number 5. This reference should be completed as following: J Orthop Trauma. 2019 Jan;33(1):e8-e13. doi: 10.1097/BOT.0000000000001329.

4.      Page 5-6. References have doubled numbering.

Author Response

RESPONSE TO REVIEWERS

MANUSCRIPT ID medicina-460093

Local Pro- and Anti-Coagulation Therapy in the Plastic Surgical Patient: A Literature Review of the Evidence and Clinical Applications

Reviewer 1:

Major point:

1.      The manuscript should be more practical and interesting for researchers and surgeons. For this reason, the authors should prepare separate tables for each, method or medication presented in the manuscript. Moreover, experimental studies on animals and clinical observation should be also, presented in separate tables. Tables should present data shown in published articles representative for each method. Tables should contain the name of the first author, year of publication, item number in References, indications and contraindications for presented method, therapeutic effects, doses and duration of treatment, side effects and their frequency, as well as prophylaxis to prevent side effects. In the text of the manuscript, the authors should discuss and conclude data presented in tables.

Thank you very much for this suggestion, we have added all tables in the format as you requested.

2.      Moreover, the authors should present/calculate average cost of each method per one patient.

While we were able to address all other points described above and below, this item could not be addressed as there is no reported data on the costs of these medications in the current literature.  Thank you again for your review and suggestions, we sincerely hope you approve of our revisions.

Minor points

1.      Page 2, line 67. “The increasingly used direct oral anticoagulants appear to have similar intraoperative transfusion requirements and post-operative bleeding risks as warfarin…” What does this sentence mean? Warfarin is the oral anticoagulant, too. This sentence should be corrected.

Thank you for the comment.  We have clarified this statement:  “The increasingly-used direct oral anticoagulants (DOACs), including the three factor Xa inhibitors (apixaban, edoxaban, and rivaroxaban) and one direct thrombin inhibitor (dabigatran) appear to have similar intraoperative transfusion requirements and post-operative bleeding risks as vitamin K antagonist therapy (warfarin)… “ The goal of the sentence was to state that post-operative bleeding risk appears to be high regardless of the mechanism of the oral anticoagulant class.  We hope this has been clarified sufficiently.  Thank you again for the comment.

2.      Page 2, line 84-86. “It also appears to be effective at reducing serum volume for bladder irrigation when applied intravesically for hematuria…” What is the meaning of this sentence? It should be clarified.

Thank you for suggesting this edit; we have revised this statement in the manuscript text.

3.       Page 5, References, item number 5. This reference should be completed as following: J Orthop Trauma. 2019 Jan;33(1):e8-e13. doi: 10.1097/BOT.0000000000001329.

Thank you for this comment. The reference you cite has been updated according to your citation. Thank you for your help.

4.      Page 5-6. References have doubled numbering.

Thank you for this comment, we have repaired the reference numbering.

Reviewer 2 Report

The review is well organized and completely described. I have no any doubt about the recommendation to accept it in present form. 

Author Response

Reviewer 2:

The review is well organized and completely described. I have no any doubt about the recommendation to accept it in present form. 

Thank you very much for your thoughtful and thorough review!  We very much hope to see this manuscript published in Medicina soon. 

Round 2

Reviewer 1 Report

Manuscript ID: medicina-460093

Title: Local Pro- and Anti-Coagulation Therapy in the Plastic Surgery Patient: A Literature Review of the Evidence and Clinical Application

Authors: Oliver JD, DeLoughery EP, Boczar D, Sisti A, Huayllani MT, Restrepo DJ, Hu MS, Forte AJ.

The second review.

The title of the manuscript is ‘Local Pro- and Anti-Coagulation Therapy in the Plastic Surgery Patient: A Literature Review of the Evidence and Clinical Application. Moreover, the authors stated that that “the aim of the review was to provide a comprehensive overview of the clinical application of topical anti-and procoagulant therapy relevant to plastic and reconstructive surgery. Those authors’ statements indicate that their review should present clinical observations on the use of pro-coagulants and anti-coagulants in plastic and reconstructive surgery. However, the manuscript by Oliver et al. is mainly based on previous review articles. In addition, the manuscript resembles an extended abstract, even information in Wikipedia is characterized by more details.

On the over hand, the search in PubMed allows to find 46 articles for tranexamic acid plastic surgery; 585 articles for tranexamic acid reconstructive surgery, 17 articles for topical thrombin plastic surgery, 38 articles for topical thrombin reconstructive surgery; 172 articles leech reconstructive surgery; 182 articles for leech plastic surgery, etc. This indicates that there are original articles on this subject. For this reason, the reviewer suggested in the previous review, that authors should prepare separate tables for each, method or medication presented in the manuscript.

The authors have prepared only two tables, and the form of these tables and data presentation are not acceptable. As was mentioned above, a separate table should be prepared for each, method or medication presented in the manuscript. Moreover, each table should include 4-5 original articles, name of the first author, year of publication, item number in References (all above presented in the column: article), indications for usage of the presented method, doses and duration of treatment, therapeutic and side effects.

The authors should prepare appropriate titles of figures and names of columns. Tables must be understandable without reference to the text.

Moreover, in the text of the manuscript, the authors should discuss and conclude data presented in tables! For example, they should show kind of study (prospective or retrospective, presence of control group, type of control group, frequency of side effects and presence of these side effects in primary case of surgery without pro-or anticoagulative treatment.

Correction of minor points is fine.

Author Response

Dear Reviewer,

We truly appreciate your thoughtful and thorough review of our manuscript.  We have taken substantial time to address your critiques thoroughly in this manuscript revision, including the inclusion of 3 tables formatted in the style and content you prescribed.

While the purpose of this manuscript was not perform a systematic review of all literature on this topic (rather, a summarized and practical review of local pro- and anti-coagulants utilized in common plastic surgical indications), we do feel that this manuscript is of great value to plastic surgeons in practice considering what has been utilized and potential future directions for more extensive research application. We have modified the statement on the goal and purpose of this review in order to better match our intended vision of this study.

We sincerely thank you again for your reviews. We truly hope that this manuscript revision will be acceptable for publication in Medicina, as this manuscript carries value to our specialty.

Gratefully yours.

Round 3

Reviewer 1 Report

The current version of the maniscript meets the conditions for its publication.